# Measured Rainfall Infiltration and the Infiltration Interface Effect on Double-Layer Loess Slope

Weishi Bai [1] , Rongjian Li [1],*, Junyi Pan [2], Rongjin Li [3], Lei Wang [1] and Zhengwu Yang [1]

1 Institute of Geotechnical Engineering, Xi'an University of Technology, Xi'an 710048, China; bai1059296512@163.com (W.B.)
2 Changqing Engineering Design Co., Ltd., Xi'an 710018, China
3 School of Management, Xi'an University of Architecture & Technology, Xi'an 710055, China
* Correspondence: lirongjian@xaut.edu.cn; Tel.: +86-139-9129-8231

**Abstract:** It is of great theoretical and engineering significance to carry out field rainfall tests and research on double-layer soil slopes in loess areas. Based on the developed rainfall simulation system with slow-moving injection, field rainfall tests were carried out on a natural double-layer loess slope. The characteristics of volumetric water content were monitored, and the rainfall infiltration characteristics and infiltration effect at the interface of the soil layer were analyzed by numerical simulation. The results showed the fastest infiltration at the top platform of the slope, followed by that at the upper surface of the slope, and the slowest infiltration at the lower surface of the slope during rainfall. Under various rainfall intensities, the erosion of the upper silty loess slope was greater than that of the lower clay loess slope, and the erosion patterns were quite different at the end of rainfall. During the infiltration process in the double-layer loess slope, a stagnant transition area was formed near the interface of the soil layer. The equipotential line of water content in the stagnant transition area of the upper region was roughly parallel to the slope surface, and the equipotential line in the lower region was roughly parallel to the interface of the soil layer. With an increase in rainfall intensity, the upper transition area at the interface of the soil layer continued to extend from the slope surface inward, showing the interface infiltration effect that became increasingly significant with the intensification of rainfall. The infiltration effect at the soil layer interface could provide an evaluation basis for rainfall infiltration analyses of multi-layer soil slopes.

**Keywords:** double-layer slope; field test; rainfall intensity; infiltration characteristics; infiltration effect at the interface



## 1. Introduction

The economic losses caused by loess landslides reach billions of dollars every year in northwest China [1], and rainfall is one of the main factors inducing slope instability [2,3]. Among more than 90 large landslides listed in large landslide cases in China [4], most of them were related to rainfall. The regularity of rainfall infiltration in slopes and slope instability caused by rainfall have become important research subjects.

In terms of rainfall infiltration tests on a homogeneous soil slope, Tang et al. [5] found that the rainfall infiltration depth of a loess slope was generally limited to within 3.0 m underground through in-situ monitoring and field investigation, and the authors summarized three basic conditions of landslide induced by rainfall infiltration. These included long periods of continuous rainfall, large cumulative percolating rainfall over long periods, and high rainfall intensity. Through field investigation, Wang et al. [6] showed that the occurrence of loess flow slip was closely related to infiltration depth, slope angle, slope morphology, rainfall intensity, and loess shear strength. Shu et al. [7] carried out a series of field artificial rainfall tests on a loess slope, which showed that the phenomena of rapid rise in pore water pressure and rapid decline of soil stress occurred when the loess slope slipped. Xie et al. [8] and Tu et al. [9] carried out field artificial rainfall tests to

study the rainfall infiltration characteristics of loess subgrade slopes, and pointed out that increasing the compaction degree of loess can effectively prevent the instability of loess slopes during rainfall. Sun et al. [10] also carried out field artificial rainfall tests to reveal the surface infiltration mechanism of an unsaturated loess slope, and pointed out that the depth of rainfall infiltration of a loess slope could hardly exceed 3.0 m under rainfall conditions. Zhan et al. [11] comprehensively discussed the interaction mechanism of soil and water in an unsaturated expansive soil slope through artificial rainfall tests and in-situ monitoring, and they concluded that rainfall infiltration would reduce the shear strength of expansive soil due to the reduction of effective stress and soil expansion softening after water absorption. Wu et al. [12] and Chang [13] studied the failure characteristics of loess slopes under artificial rainfall conditions through indoor flume tests, simulating the failure development process of a loess landslide and revealing the rainfall infiltration mechanisms of slopes with different slope angles under the same rainfall intensity. Zhang et al. [14] used an improved electrical resistivity tomography (ERT) method to study the characteristics of water infiltration and the slope failure process in loess landslides, which has certain significance for early identification and risk mitigation of disaster in loess landslide areas.

For homogeneous soil slopes, the above research results systematically revealed the infiltration mechanism of soil slopes under rainfall conditions. But in a large number of rainfall landslide cases, the soil layer composition of the sliding body presents various complexities [15]. As a result, the failure characteristics of multi-layer soil slopes are different from those of homogeneous soil slopes. In particular, the rainwater infiltration and failure mechanisms of multi-layer soil slopes are still not clear.

In terms of rainfall infiltration analysis of a heterogeneous slope, Hu et al. [16] proposed an analytical method for calculating the stability of multi-layer soil slopes during rainfall infiltration based on the unsaturated soil theory, and they obtained the infiltration depth of rainwater in a multi-layer soil slope under different rainfall conditions. Sun et al. [17] established the relationship equation between rainfall parameters and infiltration depth based on the concept of the permeability coefficient ratio of multi-layer soil, and they found that the steep increase in pore water pressure at the soil layer interface had a great influence on the instability of multi-layer soil slopes.

In terms of numerical simulation research on rainfall infiltration of a heterogeneous slope, Wang [18] built a hydrological model for a gentle slope with a dual geological structure, concluding that when the initial groundwater level and rainfall intensity were both high, the temporary confined water formed quickly, increasing the probability of landslide disaster. Ren et al. [19] carried out a numerical simulation study on a dual-structure slope with silty clay at the top and mudstone at the bottom. The study showed that the expansion of silty clay caused by rainfall and the shrinkage cracking caused by evaporation were the main factors affecting slope failure.

Combined with theoretical analysis and numerical simulation of the rainfall infiltration of a heterogeneous slope, Han et al. [20,21] analyzed the rainfall infiltration characteristics of a double-layer soil slope with upper soil and lower bedrock, and the research results showed that the accumulation and rise in pore water pressure at the soil–rock interface was the main factor in landslide failure.

The above research on the theoretical analysis and numerical simulation of the rainfall infiltration of multi-layer slopes has promoted research on the rainfall infiltration mechanism of heterogeneous slopes. However, in the field of testing, there is still a lack of research on field rainfall tests of multi-layer soil slopes. Therefore, carrying out field rainfall infiltration tests of multi-layer soil slopes is urgently needed to more intuitively study the influence of the soil layer interface characteristics of multi-layer soil slopes and reveal the rainfall infiltration mechanism of multi-layer soil slopes.

For a double-layer loess slope with silty loess in the upper layer and clay loess in the lower layer, a new rainfall simulation system with moving and sweeping-spraying raindrops [22] was used to carry out field rainfall tests under different rainfall intensity conditions, and we studied the rainwater infiltration mechanism of a double-layer loess

slope and the influence of the permeability characteristics of the soil layer interface in order to provide an analytical basis for the prevention and control of rainfall disasters of multi-layer soil slopes.

## 2. Test Site and Soil

The test site was located on the bottom slope of the ditch in Luojiagou Village, Mubo Town, Huanxian County, Gansu Province, China. The rainfall in this area is concentrated from July to September, accounting for about 60% of the annual rainfall.

At the test site, the surface of the slope was excavated manually, and then the slope was cut according to the slope ratio of 1:0.75. After excavation, the slope test site was 5.0 m long, 2.5 m wide, and 4.0 m high. Silty loess and clay loess were distributed from top to bottom, and the dividing line was roughly 2.5 m below the slope top (Figure 1). The dry densities of silty loess and clay loess were 1.55 and 1.60 g/cm$^3$, the plasticity indexes were 9.8 and 13.0, and the permeability coefficients were 2.10 $\times$ 10$^{-6}$ and 1.70 $\times$ 10$^{-7}$ cm/s, respectively. It can be seen that the permeability of the upper silty loess was greater than that of the lower clay loess.

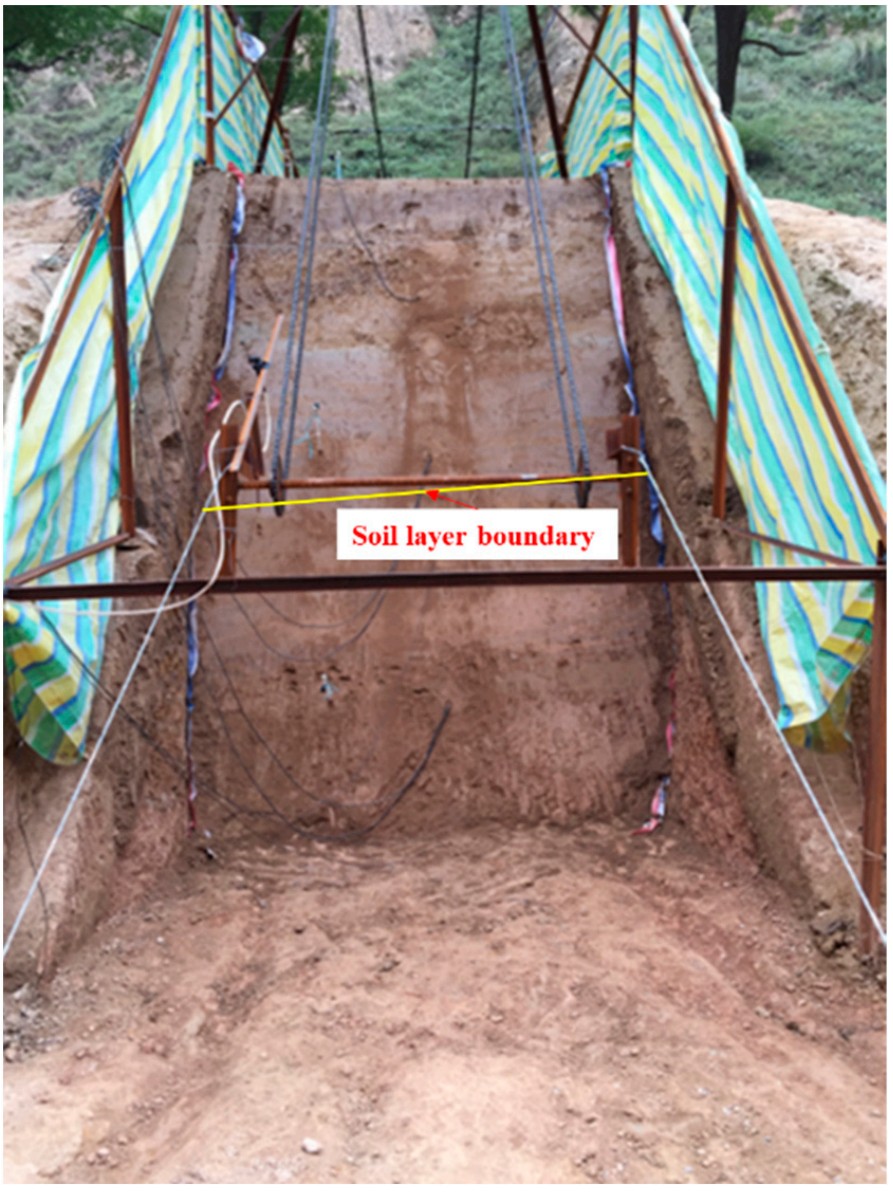

**Figure 1.** Schematic diagram of the test slope with slope cutting.

## 3. Artificial Rainfall Device and Rainfall Scheme

### 3.1. A New Rainfall Simulation System with Moving and Sweeping-Spraying Raindrops

Due to the limitations of rainfall sprinklers, it is difficult for the various equipment used in artificial rainfall simulations to simulate a variety of rainfall intensities. The main reason is that the flow of existing rainfall sprinklers is too large. In rainfall operation, only the intermittent rainfall mode can be adopted for short-term rainfall. However, this rainfall method leads to a greater degree of erosion of the slope surface, which affects the comprehensive effects of artificial rainfall simulation.

In order to better meet the requirements of rain intensity and the uniformity of natural rainfall, this research independently developed a patented technology for a new rainfall simulation system with moving and sweeping-spraying raindrops. The rainfall simulation system is mainly composed of a support device, a moving and sweeping-spraying control device, and a waterway pipeline (Figure 2). Combined with reasonable control of rainfall duration by adjusting the chain running speed, the rainfall requirements of natural rainfall can be met. A deep-buried isolated trench [23] was also used for reference to solve the boundary constraints such as lateral infiltration of rainwater and deformation lag. In order to prevent lateral seepage loss of rainwater from affecting the test results, the isolated trench with a width of 0.1 m and depth 4 m was excavated at the boundary outside the rainfall area (Figure 3a), laid with isolated cloth, backfilled, and compacted in layers (Figure 3b,c). The trench was covered to the edge of the isolated cloth and covered with soil for protection (Figure 3d). Furthermore, the geotextile covering the surface of the isolation barrier was elevated 1–2 cm above the ground, preventing any lateral overflow of slope runoff in the test site and minimizing any downward rainwater loss.

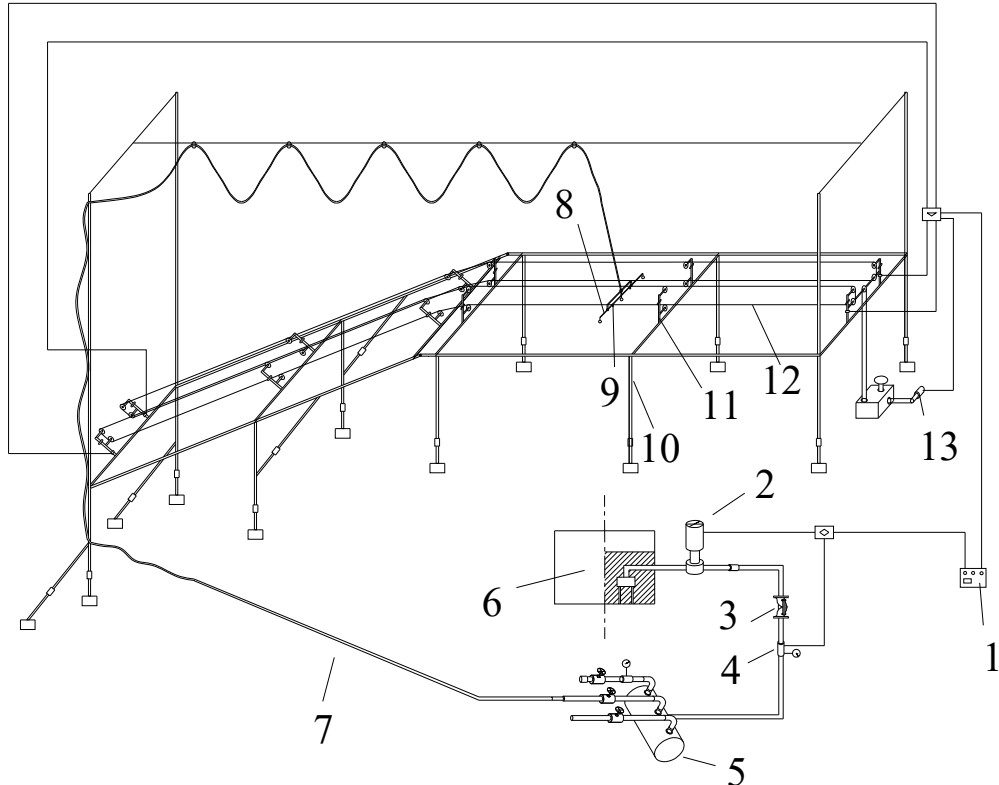

**Figure 2.** Schematic diagram of rainfall system. 1. Power supply; 2. Water pump; 3. Pressure regulating valve; 4. Pressure gauge; 5. Buffer diverter; 6. Water tank; 7. Waterway; 8. Rainfall branch pipe; 9. Rain sprinkler; 10. Rainfall support; 11. Pulleys; 12. Chain; 13. Motor.

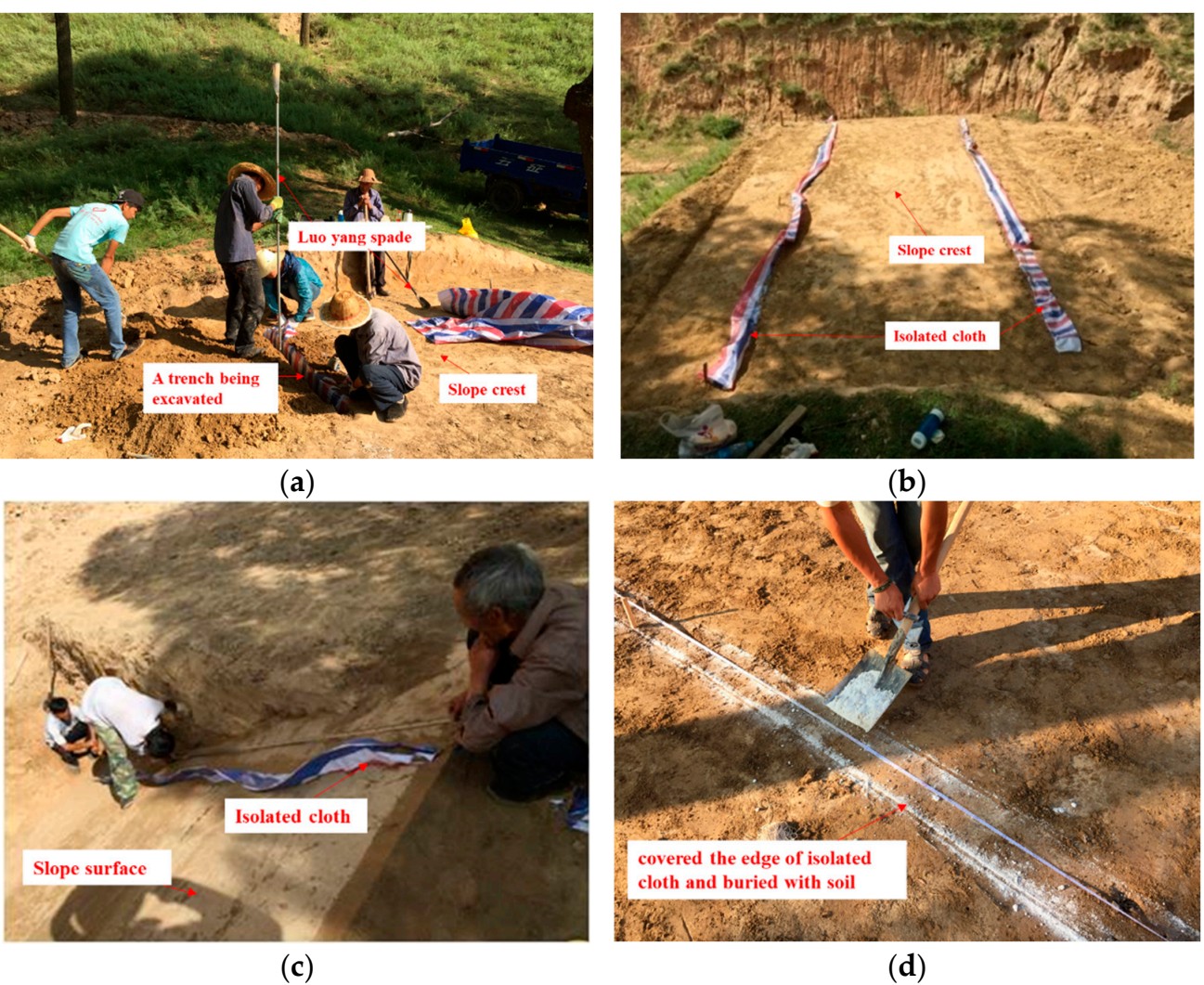

**Figure 3.** Laying deep-buried isolated trench: (**a**) excavation of isolated trench; (**b**) laying isolated cloth and backfilling soil at the slope crest; (**c**) laying isolated cloth and backfilling soil at the slope surface; (**d**) protection after construction.

In this study, a patented technology for a "moving and sweeping-spraying" slope rainfall system was independently developed. The rainfall system adopted the "sweeping and spraying" rainfall method combined with control of the circulating movement speed of the rainfall bracket to ensure that the rainfall bracket could evenly traverse each section of the test slope in a short time. Thus, the rainfall on each section of the slope to be tested was generally equivalent to the rainfall requirements of full-coverage natural rainfall.

In the rainfall system, two symmetrical down-jet fan-shaped nozzles (Figure 4) were selected, and the water outlet direction of the two nozzles was uniformly adjusted to a straight line. When the spacing between the two sprinkler was set at 1.2 m, the calibration results of the on-site rainfall were optimal, which met the requirements of a minimum uniformity of 80% during rainfall and effectively simulated different rainfall intensity conditions. This process ensured complete coverage of the slope top and surface. The nozzle calibration process is shown in Figure 4c, and the nozzle calibration results are shown in Table 1.

**Table 1.** Numerical results of nozzle calibration.

| Measuring Tube Number | 1 | 2 | 3 | 4 | 5 | 6 | 7 | 8 | 9 | 10 | 11 | 12 | 13 | 14 |
|---|---|---|---|---|---|---|---|---|---|---|---|---|---|---|
| specific water yield (mL) | 8.4 | 8.3 | 8.1 | 7.9 | 7.6 | 7.3 | 7.0 | 6.7 | 6.3 | 5.9 | 5.5 | 5.1 | 4.7 | 4.3 |

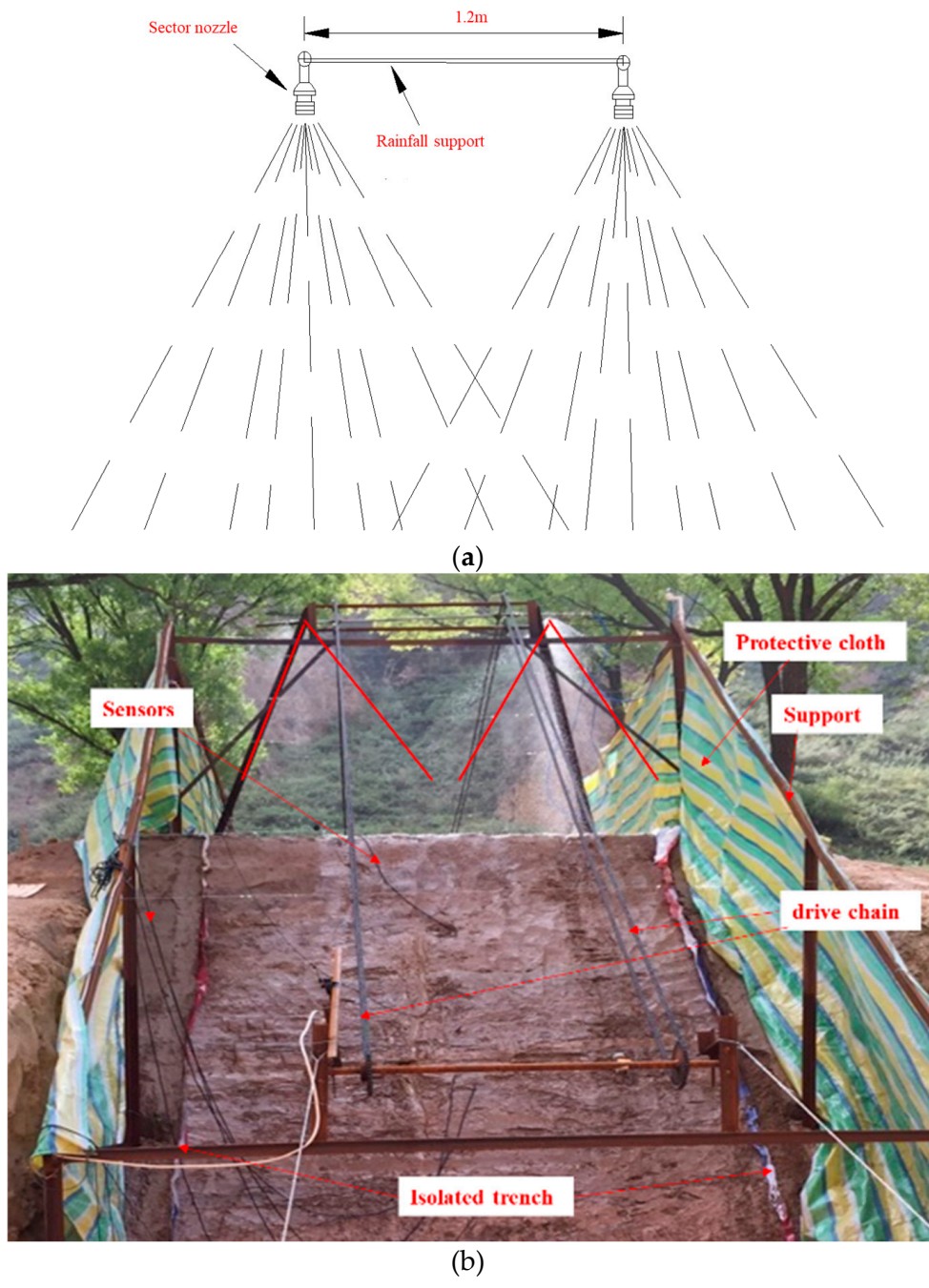

(a)

(b)

**Figure 4.** *Cont.*

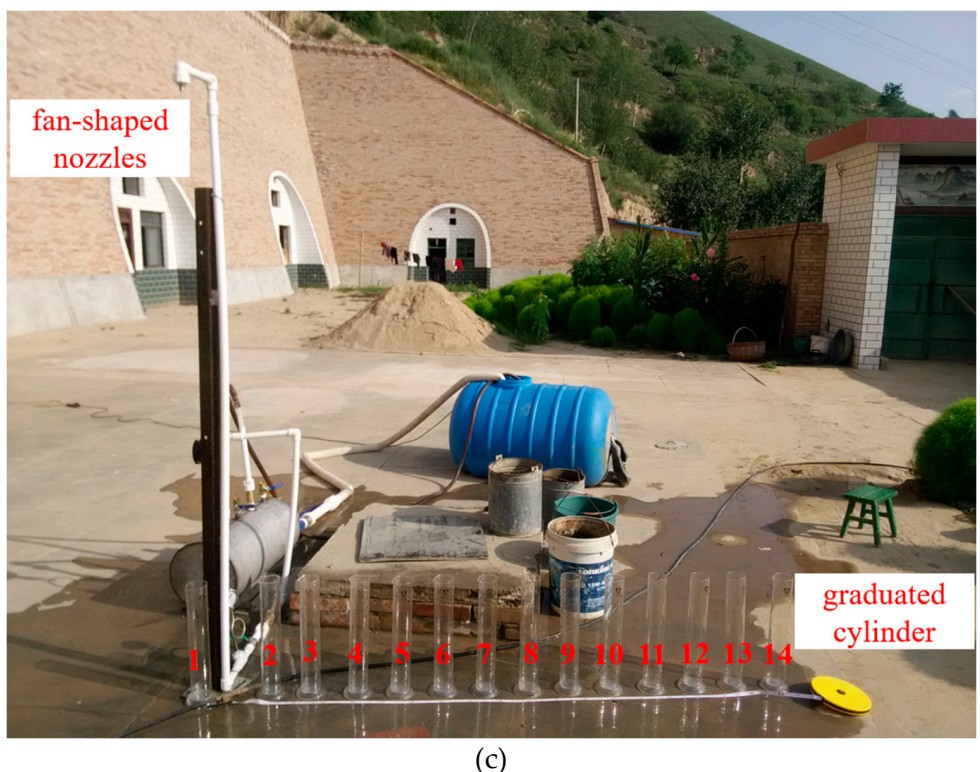

(c)

**Figure 4.** Effective rainfall from the rainfall system with moving and sweeping-spraying raindrops: (**a**) overlay effect of rain curtain; (**b**) overlay effect of rain curtain in test; (**c**) nozzle calibration schematic diagram.

### 3.2. Rainfall Test Scheme

In Huanxian County, Gansu Province, China, the average annual rainfall is 430 mm and the daily average maximum rainfall is 51.1 mm, as shown in Table 2. The average evaporation over many years is 1694.6 mm, as shown in Table 3, equivalent to a maximum daily evaporation of 4.7 mm in 24 h, so the daily evaporation in this study was selected as 5.0 mm in 24 h. The local area is in a temperate arid and semi-arid area in which rainfall is scarce, but the rainfall is concentrated and rainstorms are heavy, which easily induces serious disasters.

**Table 2.** Statistical table of average monthly rainfall in Huanxian County over the years.

| Station | Average Monthly Rainfall over the Years/mm | | | | | | | | | | | | Average Rainfall for Many Years/mm | Average Daily Maximum Rainfall/mm |
|---|---|---|---|---|---|---|---|---|---|---|---|---|---|---|
| | 1 | 2 | 3 | 4 | 5 | 6 | 7 | 8 | 9 | 10 | 11 | 12 | | |
| Huanxian County | 3 | 4 | 12 | 24 | 42 | 48 | 91 | 97 | 66 | 31 | 10 | 2 | 430 | 51.1 |

**Table 3.** Statistical table of average monthly evaporation in Huanxian County over the years.

| Station | Average Monthly Evaporation over the Years/mm | | | | | | | | | | | | Average Evaporation for Many Years/mm |
|---|---|---|---|---|---|---|---|---|---|---|---|---|---|
| | 1 | 2 | 3 | 4 | 5 | 6 | 7 | 8 | 9 | 10 | 11 | 12 | |
| Huanxian County | 57.5 | 68.9 | 125.0 | 177.1 | 252.9 | 284.6 | 213.5 | 207.2 | 120.1 | 85.2 | 57.7 | 44.9 | 1694.6 |

Moderate rain with a rainfall intensity of 28 mm in 24 h, heavy rain with a rainfall intensity of 43 mm in 24 h, and a rainstorm with a rainfall intensity of 65 mm in 24 h were designed. During the experiment, the rainfall intensity was the sum of the set rainfall

intensity and the maximum daily evaporation, that is, the rainfall intensity for moderate rain intensity was 33 mm/24 h, for heavy rain intensity it was 48 mm/24 h, and for rainstorm intensity it was 70 mm/24 h. At the same time, a 1.2 m high windbreak cloth was set on both sides of the rainfall slope during the test, which effectively avoided the rainfall loss caused by crosswind during the rainfall. Based on the double-layer loess slope with a slope ratio of 1:0.75, the whole rainfall monitoring process of this field test was divided into 24 h of rainfall monitoring and 12 h of post-rain monitoring. The whole rainfall monitoring period was from the beginning of rainfall to 12 h after rainfall, for a total of 36 h, and the data collection interval was 1 h.

*3.3. Monitoring Sensor Arrangement*

In order to monitor the change process of volumetric water content in the slope during rainwater seepage, a total of six moisture sensors were buried. The buried positions of the water sensors under three rainfall conditions are shown in Figure 5.

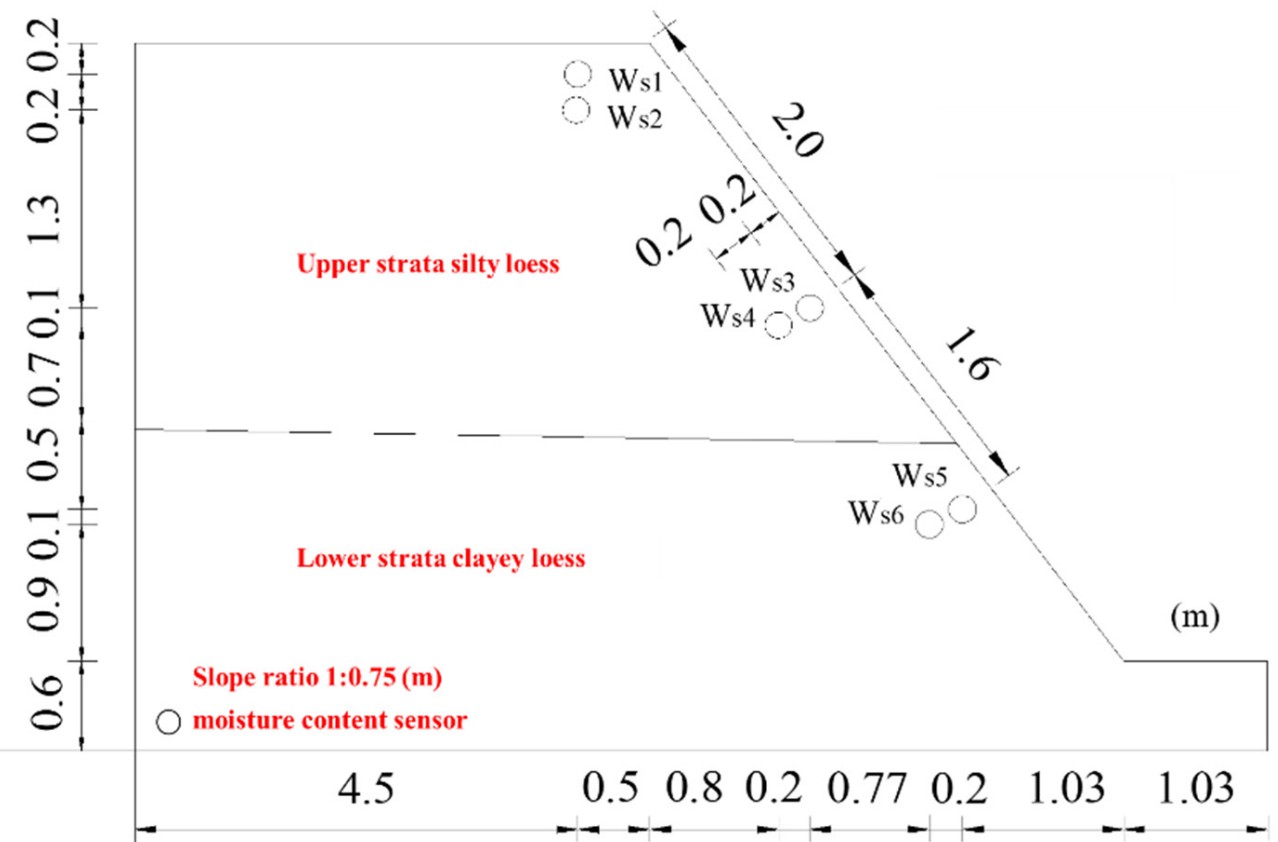

**Figure 5.** Arrangement of moisture sensors.

## 4. Analysis and Comparison of Test Results

*4.1. Test Analysis under Moderate Rain Intensity*

Figure 6 shows the slope morphology of the double-layer loess slope after 24 h of continuous rainfall under moderate rain intensity. It can be seen that at the beginning of the rainfall, the silty loess slope surface on the upper side of the slope was scoured to form fine grooves and discontinuous patchy ridges. Subsequently, the runoff phenomenon on the slope surface became more and more significant, and the spot-shaped ridges gradually eroded through into a small gully. The scouring effect of the rainwater was further intensified, leading to partial shedding of the surface of the upper silty loess slope.

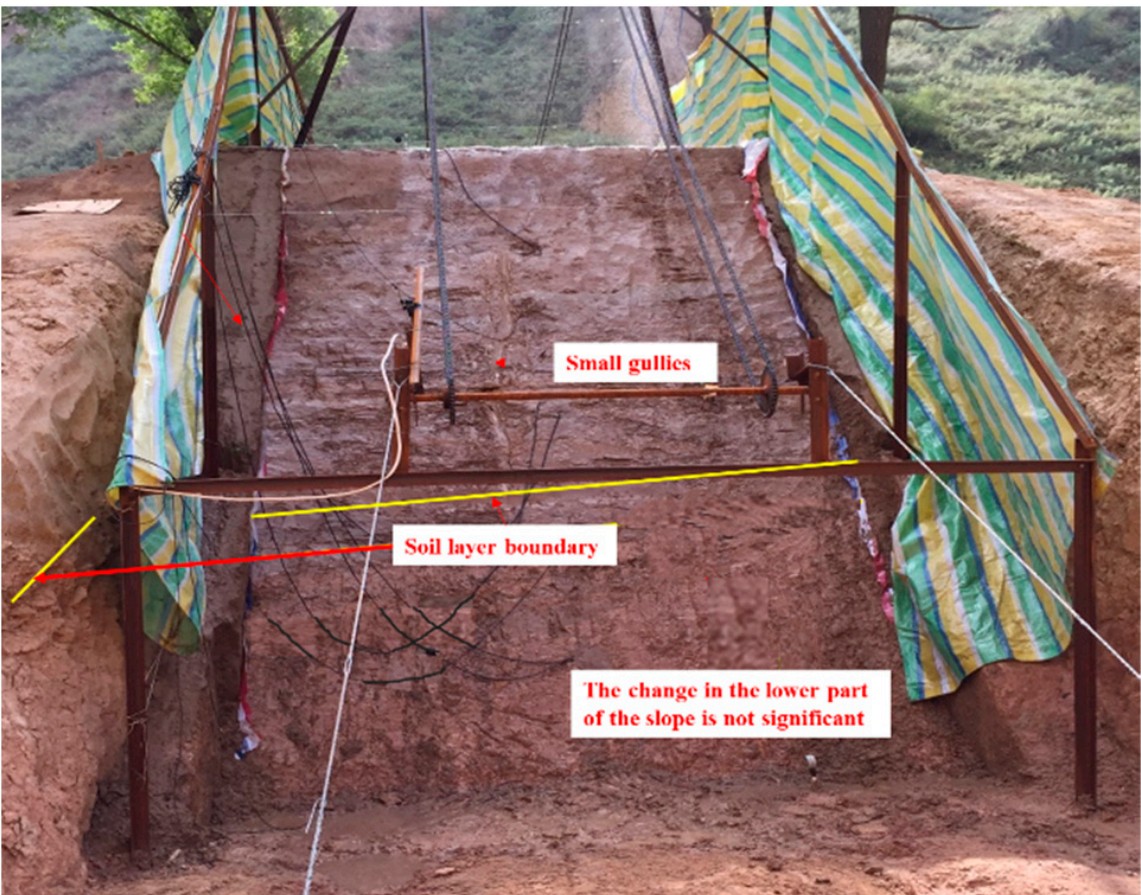

**Figure 6.** Development of slope surface under moderate rain intensity.

In addition, the soil strength of the clay loess in the lower part of the slope was higher than that of the silty loess in the upper part of the slope, and the permeability coefficient was smaller. The infiltration rate of the rainwater was relatively slow in the early rainfall and there was no obvious change in this part of the slope; the interface between the upper and lower soil layers was clearly visible after the rainfall.

It is worth noting that the double-layer soil slope presented the phenomenon of the degree of erosion of the upper silty loess slope being much greater than that of the lower clay loess slope under moderate rain intensity, and the erosion characteristics of the double-layer soil slope were obviously different from those of the homogeneous soil slope [9,12].

Figure 7 shows the change curve of moisture content at measuring points under moderate rain intensity. As can be seen from Figure 7, with continuous rainfall, the moisture content increased at measuring points in different parts of the double-layer loess slope. The moisture content at four measuring points, $W_{s1}$–$W_{s4}$, in the upper silty loess showed a trend of continuous rise at first and a trend of slight decline after the rainfall, while the moisture content at two measuring points in the lower clay loess showed a trend of continuous rise. The results showed that the upper silty loess with high permeability was positively correlated with the rainfall process, while the lower clay loess with lower permeability showed a strong lag. Secondly, the moisture content at $W_{s1}$ at the top of the slope and $W_{s3}$ in the upper silty loess soil layer with a buried depth of 20.0 cm almost changed synchronously, and the moisture content at $W_{s2}$ at the depth of 40.0 cm from the top of the slope increased much more than that at $W_{s4}$ at the depth of 40.0 cm from the slope surface, indicating that the the rainwater infiltration efficiency at the top of the slope was higher than that at the slope surface.

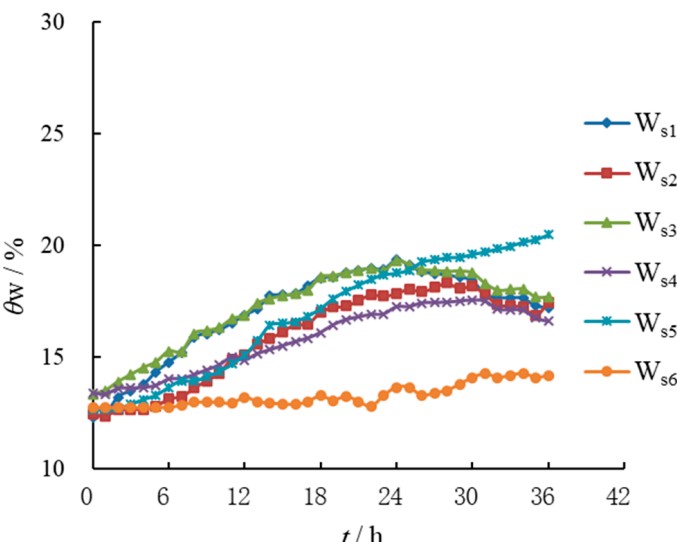

**Figure 7.** Curves of volumetric water content at different depths under moderate rain intensity.

It is worth noting that the initial growth time of water content at measuring points $W_{s5}$ and $W_{s6}$ in the lower clay loess layer was obviously lower than that at other measuring points in the upper soil, and the growth rate of water content at $W_{s6}$ was small. The reasons for these phenomena were as follows: firstly, because the permeability of the lower clay loess was significantly lower than that of the upper silty loess, the moisture content at the measuring points in the lower soil lagged behind that at the measuring points in the upper soil. Secondly, the lower soil slope infiltration wetting front was smaller, so the moisture content at the deep measuring point in the lower soil was less increased.

*4.2. Test Analysis under Heavy Rain Intensity*

Figure 8 shows the slope morphology of the double-layer loess slope after 24 h of continuous rainfall under heavy rain intensity. It can be seen that the runoff phenomenon on the slope surface was more significant, the erosion and scouring effect of rainfall on the slope surface was enhanced, the upper silty loess slope showed the local spalling mud phenomenon, the boundary between the upper and lower soil layers was covered by mud, and a small accumulative body formed in the lower clay loess area, accompanied by an obvious gully. The slope surface scouring effect under heavy rain intensity was more intense than under moderate rain intensity.

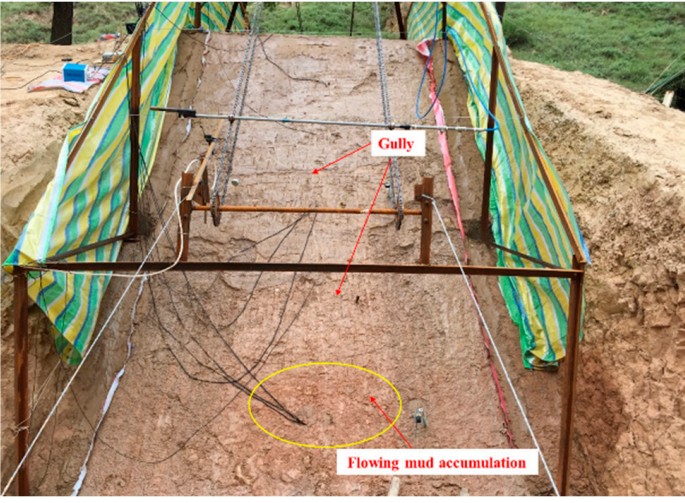

**Figure 8.** Development of slope surface under heavy rain intensity.

Figure 9 shows the change curve of moisture content at measuring points under heavy rain intensity. It can be seen that the growth rate and range of volumetric water content at each measuring point in the double-layer loess slope under heavy rain intensity were higher than under moderate rain intensity. The difference in moisture content increase between $W_{s5}$ near the slope surface in the lower clay loess and $W_{s3}$ near the slope surface in the upper silty loess was further enlarged, indicating that the infiltration difference between the different soil layers of the double-layer loess slope was intensified with the increase in rain intensity.

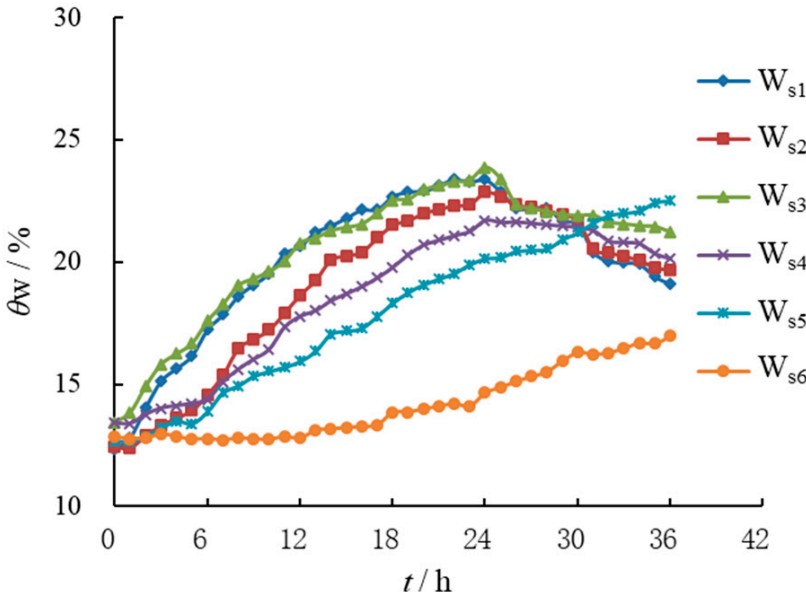

**Figure 9.** Curves of volumetric water content at different depths under heavy rain intensity.

In addition, with the increase in rainfall intensity, the infiltration wetting front of the lower soil slope slightly increased, which made the moisture content at $W_{s6}$, the deep measuring point in the lower soil, increase more than under moderate rain intensity.

*4.3. Test Analysis under Rainstorm Intensity*

Figure 10 shows the slope morphology of the double-layer loess slope after 24 h of continuous rainfall under rainstorm intensity. It can be seen that a number of runoff channels formed on the upper silty loess slope, the slope surface developed into some deep gullies, and some accumulative bodies formed in the middle of the slope and at the foot of the slope. However, there was no accumulation of silt on the lower clay loess slope except for some small gullies. This was because the permeability of the lower slope was poor, and the large-scale slope runoff caused by the rainstorm pushed the surface silt to the foot of the slope, and finally, completely different scouring effects were present on the two soil layers in the double-layer loess slope.

It can be seen that, different from the small accumulative body that formed at the foot of the homogeneous soil slope, the double-layer loess slope formed a large range of accumulative bodies at the soil layer interface and the foot of the slope under the condition of rainstorm intensity. These accumulative bodies adhered to the surface of the slope and impeded the drainage of the slope, thus forming a stagnant area and causing a steep increase in pore water pressure near the soil layer interface [18,19,21,22]. This aggravated the adverse effect of rainfall on the stability of the double-layer loess slope.

Figure 11 shows the change curve of moisture content at measuring points under rainstorm intensity. It can be seen that the growth rate and range of volumetric water content at each measuring point in the double-layer loess slope were significantly higher than under heavy rain intensity.

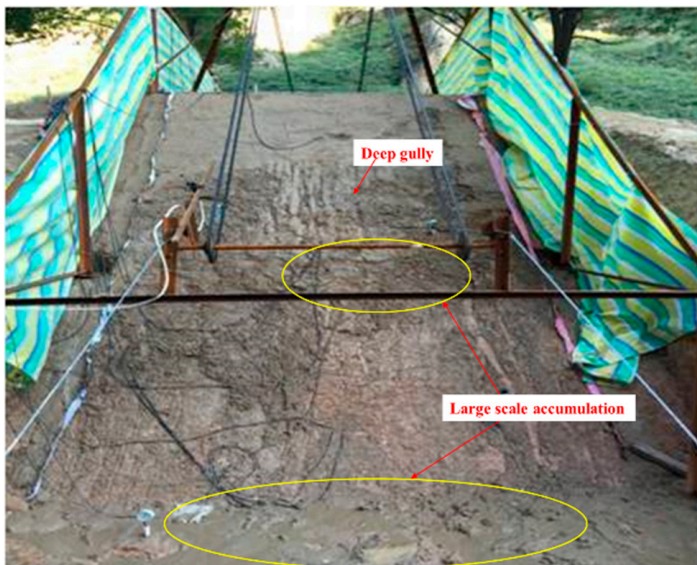

**Figure 10.** Development of slope surface under rainstorm intensity.

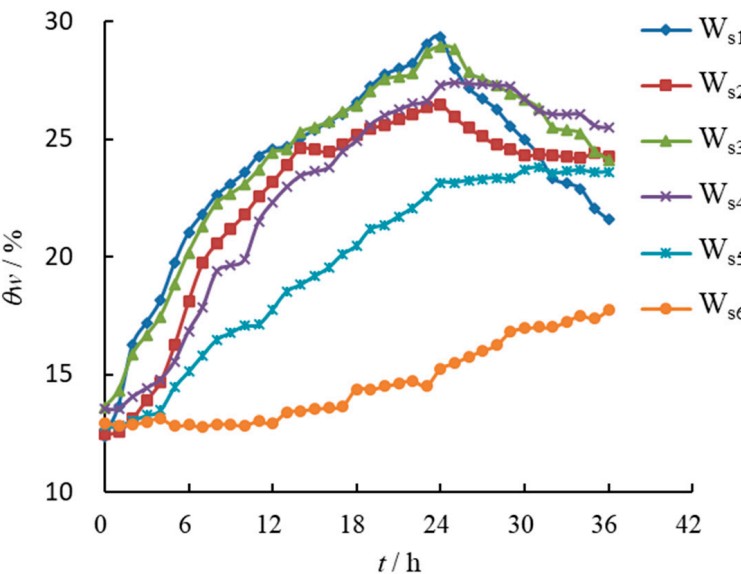

**Figure 11.** Curves of volumetric water content at different depths under rainstorm intensity.

The initial growth of water content at measuring points $W_{s1}$–$W_{s4}$ in the upper silty loess layer of the double-layer loess slope increased synchronously, while the initial growth of water content at measuring points $W_{s5}$ and $W_{s6}$ in the lower clay loess layer had a larger time difference, and the increase in water content at the upper measuring point was much larger than that in the lower layer. These results indicated that the infiltration efficiency of the upper silty loess area was significantly higher than that of the lower clay loess area under rainstorm intensity, which indicated that the infiltration difference between different soil layers of the double-layer loess slope was aggravated with the increase in rainfall intensity.

However, it is worth noting that the moisture content increase at the measuring point Ws6, which was buried 40.0 cm away from the slope surface in the lower clay loess, was close to that under the condition of heavy rain intensity, indicating that the infiltration efficiency of rainwater in the deep part of the lower clay loess slope with low permeability was still low even under heavy rain and rainstorm intensities.

## 5. Rainfall Numerical Analysis of the Double-Layer Loess Slope

### 5.1. Model Establishment and Boundary Condition

In order to comprehensively analyze the influence of rainfall infiltration on the double-layer loess slope, a two-dimensional numerical model was established in this section to study the rainfall response of the slope (Figure 12), including the influence of rainfall on the seepage field of the slope and the water distribution at the soil layer interface. The slope model was established according to the dimension information of the field test slope, and the monitoring points were set in the numerical model according to the buried positions of the moisture sensors in the field slope. The initial water content distribution in the numerical model of slope is shown in Figure 13.

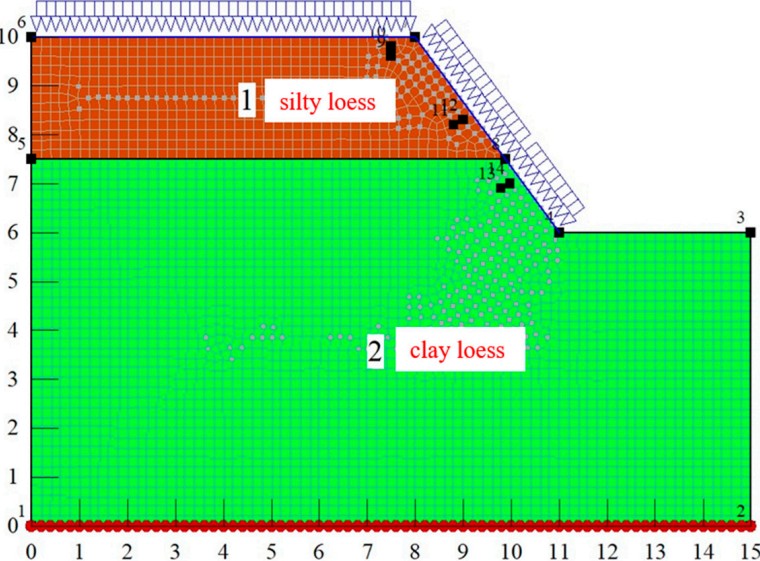

**Figure 12.** Numerical model and boundary conditions.

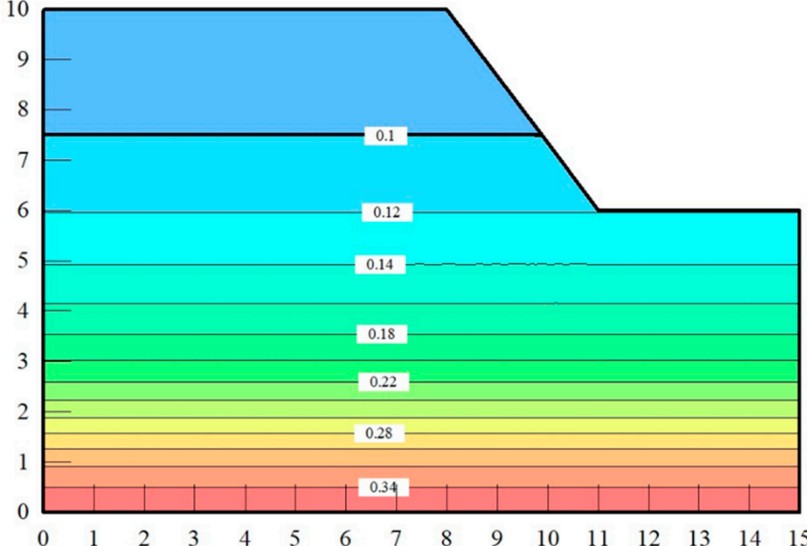

**Figure 13.** Distribution of initial water content.

In terms of boundary conditions, the flow boundary was set at the top and surface of the slope and the far-field boundary was set at the end far away from the slope infiltration. Three kinds of rain intensity were set as in the field test along with continuous rain for 24 h, and the bottom of the model was set as the zero pressure boundary.

In order to calculate and analyze the seepage mechanism of the unsaturated loess in the numerical simulation, the V-G model (Van, 1980) was selected in this study to speculate the volumetric water content function, and its governing equation is as follows:

$$\theta_{w} = \theta_{r} + \frac{\theta_{s} - \theta_{r}}{\left[1 + (\psi/a)^{n}\right]^{m}} \tag{1}$$

where $\theta_{w}$ is the volumetric moisture content; $\psi$ is the matrix suction; $\theta_{s}$ is the saturated volumetric moisture content; $\theta_{r}$ is the residual volumetric moisture content; and $a$, $n$, and $m$ are curve fitting parameters, with $m = 1 - 1/n$.

In addition, the following equation was used to estimate the permeability coefficient function accordingly:

$$k_{w} = k_{s} \frac{\left[1 - a\psi^{n-1}\left(1 + (a\psi^{n})^{-m}\right)\right]^{2}}{(1 + a\psi^{n})^{m/2}} \tag{2}$$

where $k_{w}$ is the permeability coefficient and $k_{s}$ is the saturation permeability coefficient. The fitting parameters $a$, $n$ and $m$ are 13.4 kPa, 1.56, and 0.36 in the upper silt loess layer, and 20 kPa, 2, and 0.5 in the lower clay loess layer, respectively.

### 5.2. Discussion and Analysis

Figure 14 shows the change curves of volumetric water content at each measuring point under three kinds of rain intensity calculated by numerical simulation. Compared with Figures 7, 9, and 11, it can be seen that the variation in water content at each measuring point in the numerical model was basically consistent with the variation in the field measurements of water content. With the beginning of rainfall, the water content at each measuring point in the numerical model and the field measurements began to increase. After 24 h of rainfall, the water content at most measuring points reached the peak. After the rainfall ended, the water content at four measuring points, $W_{s1}$–$W_{s4}$, in the upper silty loess began to decrease, while the water content at two measuring points, $W_{s5}$–$W_{s6}$, in the lower clay loess continued to rise, indicating that the rainfall rainwater had gathered at the lower part of the slope and the foot of the slope, and the rainfall infiltration of the lower clay loess had a certain lag.

By comparing Figure 14a–c, it can be seen that the variation amplitude of volumetric water content at the same measuring point in the upper silty loess obviously increased with the increase in rainfall intensity, while the volumetric water content at the measuring point in the lower clay loess increased slightly, and the peak value of volumetric water content at the measuring point in the upper layer was much larger than that in the lower layer under rainstorm intensity. The results showed that rain intensity and soil permeability were the main factors leading to the difference in the seepage mechanism of the double-layer soil slope.

In addition, the volumetric water content at each measuring point in the upper silty loess layer had a small interval of initial growth time, showing a relatively consistent growth trend within a few hours after the rainfall began; however, the difference between the initial growth time at each measuring point in the lower clay loess layer was very large. This indicated that in the upper silty loess with a large permeability coefficient, the rainwater infiltration process varied little within a certain depth, while in the lower clay loess with a small permeability coefficient, the rainwater infiltration rate varied greatly at different depths.

The seepage field and volumetric water content distribution of the double-layer soil slope under moderate rain, heavy rain, and rainstorm conditions at 24 h of rainfall are shown in Figure 15a–c. It can be seen that with the increase in rainfall intensity, the infiltration depth of the upper silty loess increased significantly and tended to be saturated near the slope surface, while the infiltration depth of the lower clay loess was relatively small, resulting in significant differences in the soil layer interface permeability characteristics. In

the upper silty loess area, the spacing of the equipotential line of volumetric water content was larger and the gradient was smaller, while in the lower clay loess area, the spacing of the equipotential line of volumetric water content was smaller and the gradient was larger. Compared to the rainfall infiltration mechanism of a homogeneous soil slope [24], it can be seen that the rainfall permeability of the double-layer loess slope had an obvious permeability difference at the soil layer interface.

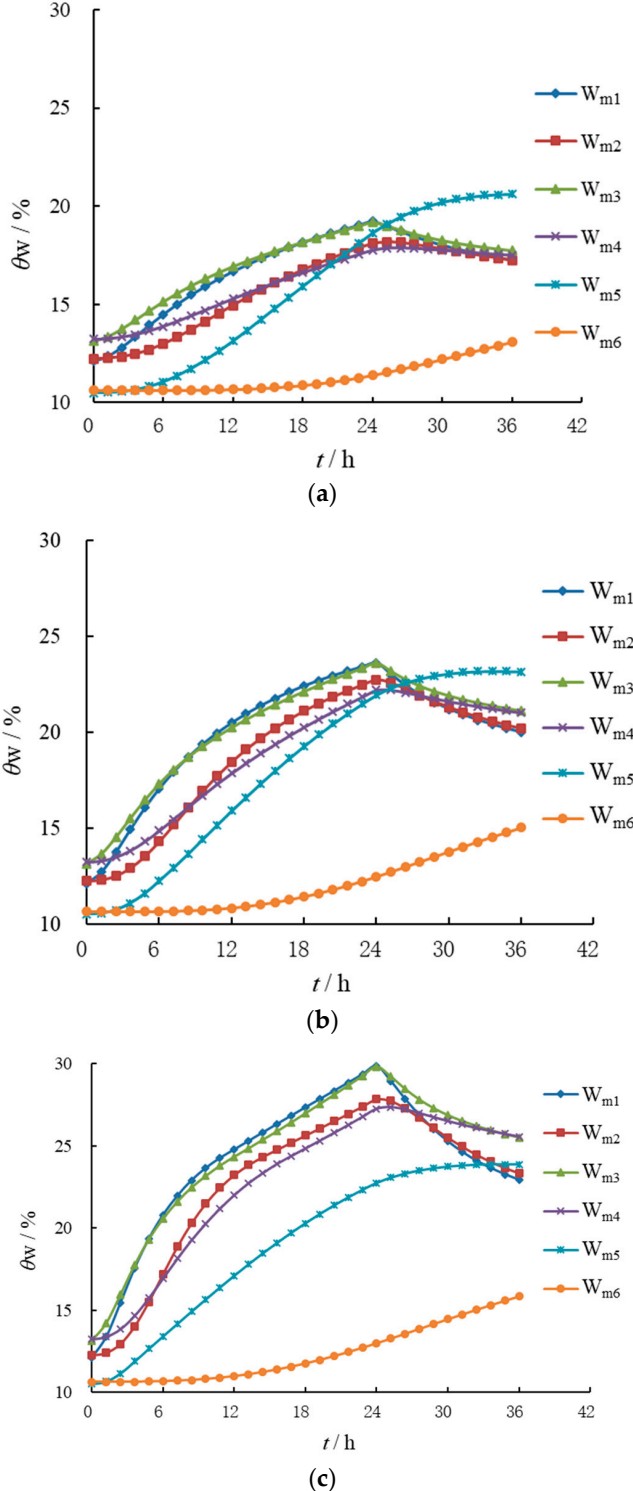

**Figure 14.** Curves of volumetric water content under different rain intensities in the numerical simulation: (**a**) moderate rain; (**b**) heavy rain; (**c**) rainstorm.

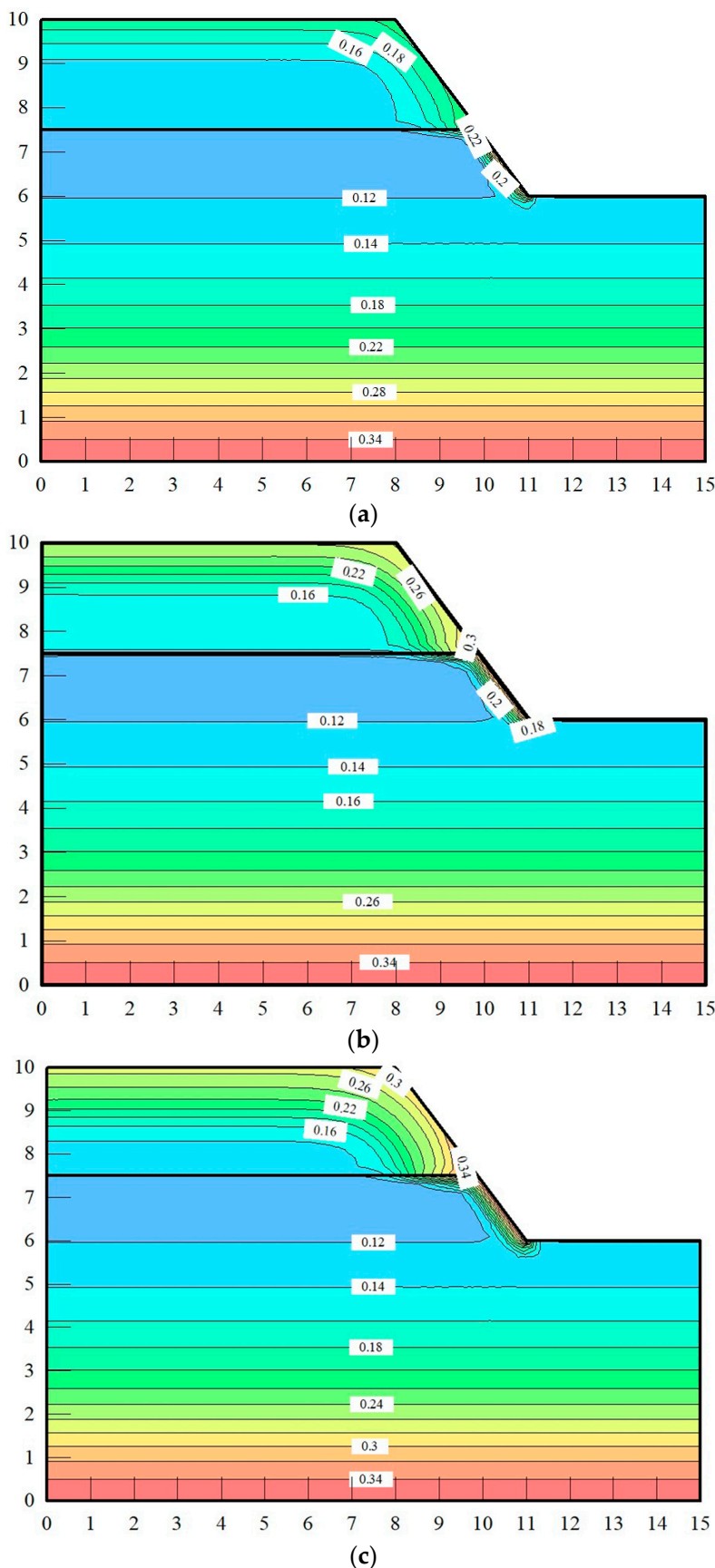

**Figure 15.** Distribution of volumetric water content at 24 h: (**a**) moderate rain at 24 h; (**b**) heavy rain at 24 h; (**c**) rainstorm at 24 h.

Through the analysis of the infiltration mechanism in Figure 15a–c, it can be seen that, with continuous rainfall, because the permeability of the upper soil of the double-layer loess slope was larger while that of the lower soil was smaller, a water-stagnant layer was formed at the double-layer loess interface during the infiltration process, so a transition area was formed near the slope surface at the soil layer interface of the two layers. Thus, the interface permeability effect was generated, that is, the equipotential line of water content in the upper region was roughly parallel to the slope surface, while that in the lower region was roughly parallel to the soil layer interface. With the increase in rainfall intensity, the upper transition area at the soil layer interface extended more violently from the slope surface to the slope body, and the permeability effect of the double-layer loess interface became more significant.

## 6. Conclusions

In this paper, three field rainfall tests were carried out for the double-layer loess slope, and the rainwater infiltration mechanism and seepage characteristics at the soil layer interface were studied. The conclusions are as follows:

Under the conditions of continuous rainfall for 24 h, the scouring morphology of the upper and lower slopes of the double-layer loess slope differed greatly during rainfall. During moderate rainfall, some fine gullies formed through the upper silty loess layers on the slope surface in the upper silty loess of the double-layer loess slopes, while the slope surface in the lower clay loess did not change. After moderate rainfall, the boundary between the upper and lower layers was clearly visible. During heavy rainfall, the slope surface in the upper silty loess produced small-scale spalling fluid mud, the slope surface in the lower clay loess runoff was obvious, and the boundary of the soil layer was covered by fluid mud. During the rainstorm, the upper slope developed into some deep gullies and the lower slope had some shallow gullies, and a large range of deposits was formed at the soil layer interface and the foot of the slope.

Under different rainfall intensities, the infiltration of rainwater in the slope was fastest in the slope crest, followed by that in the upper slope surface, and it was slowest in the lower slope surface. However, the variation amplitude of the moisture content at the same measuring point in the upper silty loess obviously increased with the increase in rainfall intensity, while the moisture content of the lower clay loess increased little, indicating that the permeability of the two soil layers and rainfall intensity were the main factors leading to the difference in the seepage mechanism of the upper and lower soil layers of the double-layer loess soil slope.

Rainwater infiltration formed a transition area at the soil layer interface near the slope surface of the double-layer loess slope and produced the interface infiltration effect, showing that the equipotential line of water content in the upper region was roughly parallel to the slope surface, while the equipotential line of water content in the lower region was roughly parallel to the soil layer interface. Additionally, with the increase in rainfall intensity, the upper transition area at the soil layer interface extended more violently from the slope surface to the slope body.

In this paper, the rainfall field tests on a double-layer loess slope well realized the change process of the seepage field with rainwater infiltration in a double-layer loess slope, which lays the necessary test analysis foundation for further study of the rainwater infiltration of multi-layer soil slopes by means of testing and theory.

**Author Contributions:** W.B., analyses and writing; R.L. (Rongjian Li), analyses; J.P., data; R.L. (Rongjin Li), data; L.W., field test; Z.Y., field test. All authors have read and agreed to the published version of the manuscript.

**Funding:** This research was supported by the National Natural Science Foundation of China (No. 12102379); the Key R&D program of Shaanxi Province (2020ZDLGY07-03); and the Yan'an Science and Technology Plan Project: 2022SLSFGG-004.

**Data Availability Statement:** The data presented in this study are all available in the article.

**Conflicts of Interest:** The authors declare no conflict of interest.

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
