# Peer review of "Measured Rainfall Infiltration and the Infiltration Interface Effect on Double-Layer Loess Slope"

_water, doi:10.3390/w15142505_

Round 1
Reviewer 1 Report
1. It is believed that the infiltration velocity or characteristics of the rainfall into the slope heavily depends on the permeability of each soil layer. Therefore, the conclusions about the infiltration speed in the top slope platform, upper and lower surfaces of the slope might be reconsidered.
2. Why the isolated trench can prevent lateral seepage loss of rainwater? It seems that this can just prevent the rainwater from escaping through lateral seepage loss. It can be lost through surface runoff.
3. Line154-156: The calibration data is suggested to be added into this part.
4. Line160: What’s the meaning of “moving slow injection”?
5. Line165: How to technically control the daily evaporation in this research?
6. Line170: Why the monitoring period is just 12 h?
7. The reviewer thinks that the rainfall should evenly cover the whole slope.
8. The physical properties of the two layers of soil should be added.
9. The numerical simulation should be compared with the test results.
10. Line405-Line412: Why the review comments are part of the manuscript?
11. The interface effect of infiltration is not deeply analyzed at present.
12. What’s the difference if the slope soil is not loess? Therefore, how to consider the special property of loess?
The English writing can be accepted based on some minor improvements.
Reviewer 2 Report
1. Title: The current title, “Infiltration in a rainfall field test on a double-layer loess slope and the interface effect of infiltration”, is obscure, it is suggested to consider “Rainfall measured infiltration and infiltration interface effect on double-layer loess slope.”
2. Assumptions: Lines 301~303, while the use of the V-G model for estimating the volumetric water content function and permeability coefficient function is reasonable, it would be beneficial for the authors to discuss any assumptions made in using this model and their potential implications for the results.
3. Boundary conditions: Line 298, the authors have used certain boundary conditions in their numerical simulations. A more detailed discussion on the selection of these conditions, as well as an exploration of how different boundary conditions could potentially impact the results, would add depth to the paper.
4. Rainfall intensities: Lines 162~170, the study investigates three types of rainfall (moderate, heavy, and rainstorm). Specific rainfall intensities were chosen to carry out experiments, but the reasons behind these choices are unclear. Please provide a more detailed justification for these choices or consider a broader range of rainfall intensities to enhance the robustness of their findings.
5. Implications and applications: While the authors have generated a lot of valuable data, they could expand on the practical implications of their findings. They could, for instance, discuss potential applications in land-use planning, civil engineering, or hazard prediction and management.
6. Scientific Writing Issues: Although the manuscript is well structured overall, with no major grammatical errors or inappropriate language, minor improvements could be made. In particular, changes to the abstract and conclusion section could enhance its impact. For example, but not limited to, the following sentences can be improved.
i) Lines 11-12: 1. "It is of great theoretical and engineering significance to carry out field rainfall test and research on the double layer soil slope in loess area." --> "It is of great theoretical and engineering significance to carry out field rainfall tests and research on the double-layer soil slope in the loess area." (Plural form "tests" is needed, "double-layer" should be hyphenated as a compound adjective, and "the" is required before "loess area".)
ii) Lines 12-13: "Based on the developed rainfall simulation system with moving slow injection, field rainfall tests were carried out on natural double layer loess slope." --> "Based on the developed rainfall simulation system with slow-moving injection, field rainfall tests were carried out on a natural double-layer loess slope." (The adjective "slow-moving" should come before "injection", "double-layer" should be hyphenated, and an article "a" is needed before "natural double-layer loess slope".)
iiii) Lines 23-26: "With the increase of rainfall intensity, the upper transition area at the interface of the soil layer continued to extend from the slope surface to the inside, showing the interface infiltration effect that became more and more significant with the increase of rain intensity." --> "With the increase in rainfall intensity, the upper transition area at the interface of the soil layer continued to extend from the slope surface inward, showing the interface infiltration effect that became increasingly significant with the intensification of rainfall." ("Increase in" is the appropriate phrasing. "Inside" should be replaced with "inward" for clarity. "More and more" could be replaced with "increasingly" for more formal and concise writing. "Increase of rain intensity" should be replaced with "intensification of rainfall".)
Author Response
请看附件。

Round 2
Reviewer 1 Report
All the questions have been well responded by the authors. Therefore, the manuscript can be considered for acceptance now.
Reviewer 2 Report
The latest version has seen steady improvement. The authors responded to all six comments, addressing four comments. Most responses have been accepted. For comment 2 and comment 5, the authors have provided further explanations for the comments not well addressed. These efforts have substantially enhanced the quality of the paper, and I believe the manuscript now meets the average academic publication level of the journal of water.
The current version is good in language quality.